# GraphMaker: Can Diffusion Models Generate Large Attributed Graphs?

## Abstract

Large-scale graphs with node attributes are fundamental in real-world scenarios, such as social and financial networks. The generation of synthetic graphs that emulate real-world ones is pivotal in graph machine learning, aiding network evolution understanding and data utility preservation when original data cannot be shared. Traditional models for graph generation suffer from limited model capacity. Recent developments in diffusion models have shown promise in merely graph structure generation or the generation of small molecular graphs with attributes. However, their applicability to large attributed graphs remains unaddressed due to challenges in capturing intricate patterns and scalability. This paper introduces GraphMaker, a novel diffusion model tailored for generating large attributed graphs. We study the diffusion models that either couple or decouple graph structure and node attribute generation to address their complex correlation. We also employ node-level conditioning and adopt a minibatch strategy for scalability. We further propose a new evaluation pipeline using models trained on generated synthetic graphs and tested on original graphs to evaluate the quality of synthetic data. Empirical evaluations on real-world datasets showcase GraphMaker's superiority in generating realistic and diverse large-attributed graphs beneficial for downstream tasks.

## 1 Introduction

Large-scale graphs, enhanced with node attributes, are prevalent in many real-world settings. For instance, in social networks, nodes often represent individuals with associated demographic features (Golder et al., 2007). Similarly, in financial networks, nodes can correspond to agents, each furnished with a variety of account details (Allen & Babus, 2009). Learning a model to generate synthetic graphs that may mimic real-world graphs is a fundamental task in graph machine learning (ML), which has a broad spectrum of applications. A graph generative model may assist network scientists to better understand the evolution of complex networks (Watts & Strogatz, 1998; Barabási & Albert, 1999; Barrat et al., 2008; Chung & Lu, 2002; Leskovec et al., 2010). In addition, releasing the generated graphs from the model to some extent preserves data utility for public usage even if the original graph data cannot be shared (Jorgensen et al., 2016; Eliáš et al., 2020). For example, ML experts may develop learning models on public synthetic graphs for domain practitioners who need a model to process their data while cannot share their data directly.

Traditional approaches primarily use statistical random graph models to generate graphs (Erdős & Rényi, 1959; Holland et al., 1983; Barabási & Albert, 2002). These models often contain very few parameters such as triangle numbers, edge densities, numbers of node communities, and degree distributions, which often suffer from limited model capacity. AGM (Pfeiffer et al., 2014) extends these models for additionally generating node attributes, but it may only handle very few attributes due to the curse of dimensionality. Recent research on deep generative models of graphs has introduced more expressive data-driven methods. Most of those works solely focus on graph structure generation (Kipf & Welling, 2016; You et al., 2018b; Bojchevski et al., 2018; Li et al., 2018; Liao et al., 2019). Other works study molecular graph generation that involves attributes, but these graphs are often small-scale and consist of tens nodes per graph and a single categorical attribute per node (Jin et al., 2018; Liu et al., 2018; You et al., 2018a; De Cao & Kipf, 2018).

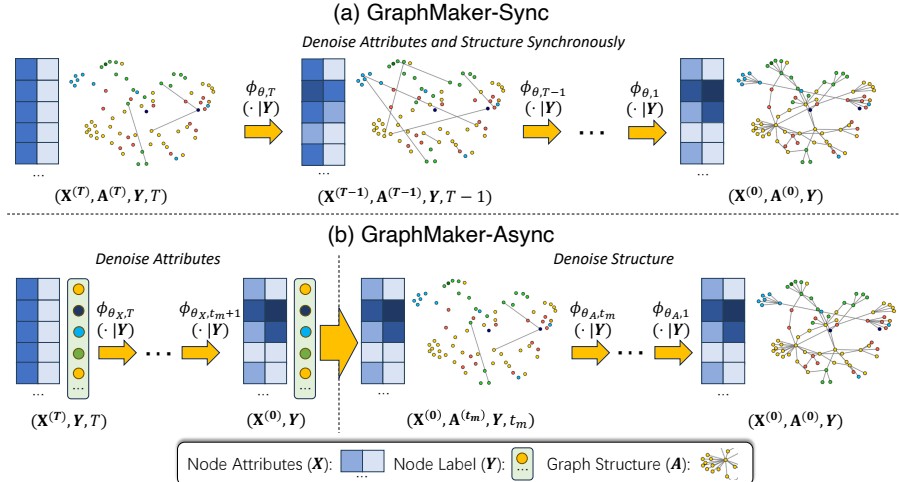

Figure 1: Generation process with two GraphMaker variants.

Diffusion models have achieved remarkable success in image generation (Ho et al., 2020; Rombach et al., 2022) by learning a model to progressively denoise a noisy sample. They have been recently extended to graph and molecule generation. Niu et al. (2020) corrupts real graphs by adding Gaussian noise to all entries of dense adjacency matrices. GDSS (Jo et al., 2022) extends the idea for molecule generation. DiGress (Vignac et al., 2023) addresses the discretization challenge induced by Gaussian noise. Specifically, DiGress employs D3PM (Austin et al., 2021) to edit individual categorical node and edge types. However, DiGress is limited to small molecule generation and cannot be applied to large attributed graph due to a scalability issue. EDGE (Chen et al., 2023) and GRAPHARM (Kong et al., 2023) propose to employ autoregressive diffusion models while can only generate graph structures with no attributes. Overall, all previous works on diffusion generative models cannot be applied to generate large attributed graphs.

Developing diffusion models of large-attributed graphs is challenging on several aspects. First, a large attributed graph presents substantially different patterns from molecular graphs, including much more skewed node degree distributions, complex correlations between high-dimensional node attributes and graph structure. Some graphs may contain node-level labels that govern the other graph components. Second, generating large graphs poses challenges to model scalability as the number of possible edges exhibits quadratic growth in the number of nodes, let alone the potential hundreds or thousands of attributes per node. Third, how to evaluate the quality of the generated graphs remains an open problem. Although deep models have the potential to capture more complicated data distribution and correlations, most previous studies just evaluate high-level statistics characterizing structural properties such as node degree distributions, and clustering coefficients of the generated graphs, which can be already captured well by early-day statistical models (Pfeiffer et al., 2012; Seshadhri et al., 2012a). Therefore, a finer-grained way to evaluate the generated large-attributed graphs is needed to justify the power of deep graph generative models.

Here we present GraphMaker, a diffusion model for generating large attributed graphs. First, we observe that generating graph structure and node attributes simultaneously with a diffusion model may not be always the best way to capture the patterns of either component and their joint patterns, though previous works by default adopt such an approach. Instead, we propose an alternative diffusion process that decouples the generation of the two components, which shows better performance in learning the generative models for many real-world graphs. Second, to better capture the correlation between node labels (if any) and the other components of graph data, we propose to generate node labels with their empirical distribution, and leverage node labels as conditions to learn a conditional diffusion model. To the best of our knowledge, this is the first work to build graph generative models with node-level information as the conditions. Third, for scalability challenges, we employ a minibatch strategy for structure prediction to avoid enumerating all node pairs per iteration during training, and design a new message-passing neural network (MPNN) to efficiently encode the data. Fourth, inspired by (Yoon et al., 2023), we propose to utilize ML models, including graph neural networks (GNNs) (Wu et al., 2019; Kipf & Welling, 2017; Gasteiger et al., 2019; Kipf & Welling,

2016)), by training them on the generated graphs and subsequently evaluating their performance on the original graphs to assess the quality of data generation. Note that Yoon et al. (2023) just generates ego-subgraphs (essentially trees) around nodes instead of the full large-attributed graphs like ours. They also train and evaluate ML models on either the original graph or generated subgraphs exclusively. Due to the strong discriminative power of ML models, our evaluation protocol works as a way to check fine-grained attribute-structure-label correlation, which compliments the evaluation based on high-level graph statistics.

Extensive studies on real-world networks with up to more than $13K$ nodes, $490K$ edges, and $1K$ attributes demonstrate that GraphMaker overall significantly outperforms the baselines in producing graphs with realistic properties and high utility for downstream graph ML tasks. For property evaluation, GraphMaker achieves the best performance for 2/3 cases across all datasets and metrics. For evaluation on graph ML tasks, GraphMaker achieves the best performance for 80 % cases. In addition, we demonstrate the capability of GraphMaker in generating diverse novel graphs.

## 2 PRELIMINARIES

Consider an undirected graph $G = (\mathcal{V}, \mathcal{E}, \mathbf{X})$ with the node set $\mathcal{V}$ and the edge set $\mathcal{E}$. Each node $v \in \mathcal{V}$ is associated with $F$-dimensional categorical attributes $\mathbf{X}_v$. In many large attributed graphs, nodes are additionally associated with a categorical label $\mathbf{Y} \subset [C_Y]^N$, where $C_Y$ is the number of node classes and $N = |\mathcal{V}|$. Note that we are to generate graphs without edge attributes to show a proof of concept, while the method can be extended to generate the case with edge attributes.

We aim to learn $\mathbb{P}_{\mathcal{G} \times \mathcal{Y}}$ based on one graph $(G, \mathbf{Y})$ sampled from it. In Section 3.4, we provide some reasoning on why the distribution may be possibly learned from one single observation. As $\mathbb{P}_{\mathcal{Y}}$ can be typically easily estimated from the data, we propose to learn this joint distribution via the conditional distribution $\mathbb{P}_{\mathcal{G}|\mathcal{Y}}$. Hence, our model can be denoted as a conditional generative model $\mathbb{P}_{\mathcal{G}|\mathcal{Y}}^{\theta}$. Note that conditional generative models have been proven more valuable in many applications than unconditional ones, such as generating images consistent with a target text description (Rombach et al., 2022) or material candidates that satisfy desired properties (Vignac et al., 2023), which often ask for controllable generation.

## 3 GRAPHMAKER

### 3.1 FORWARD DIFFUSION PROCESS AND REVERSE PROCESS

Our diffusion model extends D3PM (Austin et al., 2021) to large attributed graphs. There are two phases. The forward diffusion process corrupts the raw data by progressively perturbing its node attributes or edges. Let $\mathbf{X} \in \mathbb{R}^{N \times F \times C_X}$ be the one-hot encoding of the categorical node attributes, where for simplicity we assume all attributes to have $C_X$ possible classes. By treating the absence of edge as an edge type, we denote the one-hot encoding of all edges by $\mathbf{A} \in \mathbb{R}^{N \times N \times 2}$. Let $G^{(0)}$ be the real graph data $(\mathbf{X}, \mathbf{A})$.

We individually corrupt node attributes and edges of $G^{(t-1)}$ into $G^{(t)}$ by first obtaining noisy distributions with transition matrices and then sampling from them. By composing the noise over multiple time steps, for $G^{(t)} = (\mathbf{X}^{(t)}, \mathbf{A}^{(t)})$ at time step $t \in [T]$, we have $q(\mathbf{X}_{v,f}^{(t)}|\mathbf{X}_{v,f}) = \mathbf{X}_{v,f}\bar{\mathbf{Q}}_{X_f}^{(t)}$ for any $v \in [N]$, $f \in [F]$ and $q(\mathbf{A}^{(t)}|\mathbf{A}) = \mathbf{A}\bar{\mathbf{Q}}_A^{(t)}$. To ensure $G^{(t)}$ to be undirected, $\bar{\mathbf{Q}}_A^{(t)}$ is only applied to the upper triangular part of $\mathbf{A}$. The adjacency matrix of $G^{(t)}$, i.e., $\mathbf{A}^{(t)}$, is then constructed by symmetrizing the matrix after sampling. We consider a general formulation of $\bar{\mathbf{Q}}_{X_f}^{(t)}$ and $\bar{\mathbf{Q}}_A^{(t)}$ that allows *asynchronous* corruption of node attributes and edges. Let $\mathcal{T}_X = \{t_X^1, \cdots, t_X^{T_X}\} \subset [T]$ be the time steps of corrupting node attributes, and $\mathcal{T}_A = \{t_A^1, \cdots, t_A^{T_A}\} \subset [T]$ be the time steps of corrupting edges. Then,

$$\bar{\mathbf{Q}}_{X_f}^{(t)} = \bar{\alpha}_{\gamma_X(t)}\mathbf{I} + \left(1 - \bar{\alpha}_{\gamma_X(t)}\right)\mathbf{1}\mathbf{m}_{X_f}^{\top}, \quad \bar{\mathbf{Q}}_A^{(t)} = \bar{\alpha}_{\gamma_A(t)}\mathbf{I} + \left(1 - \bar{\alpha}_{\gamma_A(t)}\right)\mathbf{1}\mathbf{m}_A^{\top} \quad (1)$$

where $\bar{\alpha}_{\gamma_Z(t)} = \bar{\alpha}_{|\{t_Z^i|t_Z^i \leq t\}|}$ if $t \leq t_Z^{T_Z}$ or otherwise $\bar{\alpha}_{\gamma_Z(t)} = \bar{\alpha}_0 = 1$, for $Z \in \{X, A\}$. We consider the popular cosine schedule in this paper (Nichol & Dhariwal, 2021), where $\bar{\alpha}_{\gamma_Z(t)} = $

$\cos^2(\frac{\pi}{2}\frac{\gamma_Z(t)/|\mathcal{T}_Z|+s}{1+s})$ for some small $s$, but the model works for other schedules as well. $\mathbf{I}$ is the identity matrix, $\mathbf{1}$ is a one-valued vector, $\mathbf{m}_{X_f}$ is the empirical marginal distribution of the $f$-th node attribute in the real graph, $\mathbf{m}_A$ is the empirical marginal distribution of the edge existence in the real graph, and $^\top$ denotes transpose.

During the reverse process, the second phase of the diffusion model, a denoising network $\phi_{\theta,t}$ is trained to perform one-step denoising $p_\theta(G^{(t-1)}|G^{(t)}, \mathbf{Y}, t)$, where node labels are employed for guidance. Once trained, we can iteratively apply this denoising network to a noisy graph sampled from the prior distribution $\prod_{v=1}^{\hat{N}}\prod_{f=1}^{F}\mathbf{m}_{X_f}\prod_{1\leq u\leq \hat{N}}\prod_{u<v\leq \hat{N}}\mathbf{m}_A$ for data generation. While we consider $\hat{N} = N$ in this paper, the model is capable of generating graphs of a size different from the original graph. We model $p_\theta(G^{(t-1)}|G^{(t)}, \mathbf{Y}, t)$ as a product of conditionally independent distributions over node attributes and edges.

$$p_\theta(G^{(t-1)}|G^{(t)}, \mathbf{Y}, t) = \prod_{v=1}^{N}\prod_{f=1}^{F} p_\theta(\mathbf{X}_{v,f}^{(t-1)}|G^{(t)}, \mathbf{Y}, t) \prod_{1\leq u<v\leq N} p_\theta(\mathbf{A}_{u,v}^{(t-1)}|G^{(t)}, \mathbf{Y}, t) \quad (2)$$

Sohl-Dickstein et al. (2015) and Song & Ermon (2019) show that we can train the denoising network to predict $G^{(0)}$ instead of $G^{(t-1)}$ as long as $\int q(G^{(t-1)}|G^{(t)}, t, G^{(0)})dp_\theta(G^{(0)}|G^{(t)}, \mathbf{Y}, t)$ has a closed-form expression. This holds for our case as detailed in Appendix A.

## 3.2 TWO INSTANTIATIONS

To model the complex correlations between node attributes and graph structure, we study two particular instantiations of GraphMaker, named GraphMaker-Sync and GraphMaker-Async.

**GraphMaker-Sync.** GraphMaker-Sync employs a forward diffusion process that simultaneously corrupts node attributes and edges for all time steps, which corresponds to setting $\mathcal{T}_X = \mathcal{T}_A = [T]$. The denoising network is trained to recover clean node attributes and edges from corrupted node attributes and edges. During generation, it first samples clean node labels $\mathbf{Y}$, noisy attributes $\mathbf{X}^{(T)}$, and noisy edges $\mathbf{A}^{(T)}$ from the prior distributions. It then repeatedly invokes the denoising network $\phi_{\theta,t}$ to predict $G^{(0)}$ for computing the posterior distribution $p_\theta(G^{(t-1)}|G^{(t)}, \mathbf{Y}, t)$, and then samples $G^{(t-1)}$. Figure 1(a) provides an illustration.

**GraphMaker-Async.** In practice, we find that GraphMaker-Sync cannot well capture certain patterns like clustering coefficient distribution and triangle count, as shown in Section 4.2. We suspect that this problem stems from synchronous refinement of node attributes and graph structure, hence propose to denoise node attributes and graph structure asynchronously instead. We consider a simple and practically effective order as a proof of concept, which partitions $[1, T]$ into two subintervals $\mathcal{T}_A = [1, t_m]$ and $\mathcal{T}_X = [t_m + 1, T]$. The denoising network $\phi_{\theta,t}$ consists of two sub-networks. $\phi_{\theta_X,t}$ is an MLP trained to reconstruct node attributes given $(\mathbf{X}^{(t)}, \mathbf{Y}, t)$. $\phi_{\theta_A,t}$ is trained to reconstruct edges given $(\mathbf{A}^{(t)}, \mathbf{X}, \mathbf{Y}, t)$. During generation, it first samples clean node labels $Y$ and noisy attributes $\mathbf{X}^{(T)}$, then repeatedly invokes $\phi_{\theta_X,t}$ until the generation of node attributes is finished. Finally, it samples noisy edges $\mathbf{A}^{(T)}$ and invokes $\phi_{\theta_A,t}(\mathbf{A}^{(t)}, \mathbf{X}, \mathbf{Y}, t)$ repeatedly to complete the edge generation. Figure 1(b) provides a visual illustration.

## 3.3 SCALABLE DENOISING NETWORK ARCHITECTURE

Large attributed graphs consist of more than thousands of nodes. This poses severe challenges to the scalability of the denoising network. We address the challenge by improving both the encoder and the decoder parts of the denoising network. The encoder computes representations of $G^{(t)}$ and the decoder transforms them into predictions of node attributes and edges.

**Graph encoder.** To enhance the scalability of the graph encoder, we propose a message passing neural network (MPNN) with complexity $O(|\mathcal{E}|)$ instead of using a graph transformer (Dwivedi & Bresson, 2021) with complexity $O(N^2)$, which was employed by previous graph diffusion generative models (Vignac et al., 2023). Empirically, we find that with a 16-GiB GPU, we can use at most a single graph transformer layer, with four attention heads and a hidden size of 16 per attention head, for a graph with approximately two thousand nodes and one thousand attributes. Furthermore,

while in theory graph transformers surpass MPNNs in modeling long-range dependencies owing to non-local interactions, it is still debatable whether long-range dependencies are really needed for large-attributed graphs. They have not demonstrated superior performance on standard benchmarks for large attributed graphs like OGB (Hu et al., 2020).

Let $\mathbf{A}^{(t)}$ and $\mathbf{X}^{(t)}$ respectively be the one-hot encoding of the edges and node attributes for the time step $t$. The encoder of the denoising network takes $\mathbf{A}^{(t)}, \mathbf{X}^{(t)}, \mathbf{Y}$ and $t$ as input. It first uses a multilayer perceptron (MLP) to transform $\mathbf{X}^{(t)}$ and the time step $t$ into hidden representations $\mathbf{X}^{(t,0)}$ and $\mathbf{h}^{(t)}$, and initializes the node label embedding $\mathbf{Y}^{(0)}$ from $\mathbf{Y}$. It then employs multiple MPNN layers: For any node $v \in \mathcal{V}$,

$$\mathbf{X}_v^{(t,l+1)} = \sigma \left( \mathbf{W}_{T \to X}^{(l)} \mathbf{h}^{(t)} + \mathbf{b}_X^{(l)} + \sum_{u \in \mathcal{N}^{(t)}(v)} \frac{1}{\sqrt{|\mathcal{N}^{(t)}(v)||\mathcal{N}^{(t)}(u)|}} [\mathbf{X}_u^{(t,l)} \| \mathbf{Y}_u^{(l)}] \mathbf{W}_{[X,Y] \to X}^{(l)} \right)$$
(3)

$$\mathbf{Y}_v^{(l+1)} = \sigma \left( \mathbf{b}_Y^{(l)} + \sum_{u \in \mathcal{N}^{(t)}(v)} \frac{1}{\sqrt{|\mathcal{N}^{(t)}(v)||\mathcal{N}^{(t)}(u)|}} \mathbf{Y}_u^{(l)} \mathbf{W}_{Y \to Y}^{(l)} \right)$$
(4)

where $\mathbf{W}_{T \to X}^{(l)}, \mathbf{W}_{[X,Y] \to X}^{(l)}, \mathbf{W}_{Y \to Y}^{(l)}, \mathbf{b}_X^{(l)}$ and $\mathbf{b}_Y^{(l)}$ are learnable matrices and vectors. $\mathcal{N}^{(t)}(v)$ consists of $v$ and the neighbors of $v$ corresponding to $\mathbf{A}^{(t)}$. $\|$ stands for concatenation. $\sigma$ consists of a ReLU layer (Jarrett et al., 2009), a LayerNorm layer (Ba et al., 2016), and a dropout layer (Srivastava et al., 2014). Different from Eq. 3, as we do not corrupt node labels, there is no need to incorporate the time-step encoding $\mathbf{h}^{(t)}$ into Eq. 4. The encoder computes the final node representations by combining the node attribute, node label, and time step representations. To improve the expressive power of the encoder, we employ the representations across all MPNN layers as in JK-Nets (Xu et al., 2018), $\mathbf{H}_v = \mathbf{X}_v^{(t,0)} \| \mathbf{X}_v^{(t,1)} \| \cdots \| \mathbf{Y}_v^{(0)} \| \mathbf{Y}_v^{(1)} \| \cdots \| \mathbf{h}^{(t)}$. We employ two separate encoders for node attribute prediction and edge prediction.

**Decoder.** The decoder performs node attribute prediction from $\mathbf{H}_v$ in the form of multi-label node classification. To predict edge existence between node $u, v \in \mathcal{V}$ in an undirected graph, the decoder performs binary node pair classification with the elementwise product $\mathbf{H}_u \odot \mathbf{H}_v$ in addition to transformations with MLPs. Due to limited GPU memory, it is intractable to perform edge prediction for all $N^2$ node pairs at once. During training, we randomly choose a subset of the node pairs for a gradient update. For graph generation, we perform edge prediction over minibatches of node pairs.

### 3.4 CAN A GRAPH GENERATIVE MODEL BE LEARNED FROM A SINGLE GRAPH?

It remains to be answered why we can learn the distribution $\mathbb{P}_{\mathcal{G}|\mathcal{Y}}$ from only one sampled graph. Due to the permutation equivariance inherent in our MPNN architecture, it becomes evident that the derived probability distribution is permutation invariant. In other words, altering the node sequence in a graph doesn't affect the model's likelihood of generating it. This suggests that our model is primarily focused on identifying common patterns across nodes instead of focusing on the unique details of individual nodes. In this case, a single, expansive graph can offer abundant learning examples—each node acting as a distinct data point.

This raises an intriguing query: Should a model for large graph generation be calibrated to recognize individual node characteristics? We think the answer really depends on the use cases. If the aim is to analyze population-level trends, individual node details might be distractions. Note that traditional graph generative models like ER (Erdős & Rényi, 1959), SBM (Holland et al., 1983), and their degree-corrected counterparts (Seshadhri et al., 2012b; Zhao et al., 2012) only capture population-level statistics, such as degree distributions and edge densities. For scenarios that involve sharing synthetic graphs with privacy considerations, omitting the node-specific details is advantageous. Conversely, if node-specific analysis is essential, our model might fall short. Overall, there's always a tradeoff between a model's capabilities, the data at hand, the desired detail level, and privacy considerations. We think of our work as a first step and will let future works dive deeper into this issue. In Section 4.5, we empirically study this point. Indeed adopting a more expressive model that employs node positional encodings and may capture some individual node details does not consistently improve the quality of the generated graphs in our evaluation.

### 3.5 CONDITIONAL GENERATION GIVEN NODE LABELS

GraphMaker is essentially a conditional generation framework, creating graphs based on provided node labels, instead of generating node labels explicitly. While it might be straightforward to treat node label as an additional node attribute, this may prove insufficient in capturing the correlation between node label and other components, as empirically demonstrated in Section 4.4.

**GraphMaker-E.** We also explore another GraphMaker variant, named GraphMaker-E, that further utilizes label conditions. This is a special case of GraphMaker-Async, which generates node attributes through an external approach conditioned on node labels. Consequently, it only requires training a denoising network for edge generation. A significant benefit of this variant is its compatibility with powerful generative models for other modalities like images and text (Rombach et al., 2022; OpenAI, 2023).

## 4 EXPERIMENTS

### 4.1 DATASETS AND BASELINES

**Datasets:** We utilize four large attributed networks for evaluation. Cora and Citeseer are citation networks depicting citation relationships among papers (Sen et al., 2008), with binary node attributes indicating the presence of keywords and node labels representing paper categories. Amazon Photo and Amazon Computer are product co-purchase networks, where two products are connected if they are frequently purchased together (Shchur et al., 2018). The node attributes indicate the presence of words in product reviews and node labels represent product categories. See Appendix B for the dataset statistics. Notably, Amazon Computer (13K nodes, 490K edges) is an order of magnitude larger than graphs adopted by previous deep generative models of graph structures (Chen et al., 2023; Kong et al., 2023), and hence it provides a challenging testbed for model scalability.

**Baselines:** For traditional baselines, we compare against Erdős–Rényi (ER) model (Erdős & Rényi, 1959) and stochastic block model (SBM) (Holland et al., 1983). For deep learning (DL) methods, we consider feature-based matrix factorization (MF) (Chen et al., 2012), graph auto-encoder (GAE) and variational graph auto-encoder (VGAE) (Kipf & Welling, 2016). Neither these baselines nor GraphMaker-E inherently possess the capability for node attribute generation. Therefore, we equip them with $p(\mathbf{Y}) \prod_v \prod_f p(\mathbf{X}_{v,f}|\mathbf{Y}_v)$ based on empirical distributions, which yields competitive model performance on node label classification using solely node attributes, as shown in Appendix D. To ensure a fair comparison, we augment the input of the DL baselines with one-hot encodings of node labels. These two extensions allow a direct comparison between GraphMaker-E and the baselines.

### 4.2 EVALUATION WITH STATISTICS

To assess the quality of the generated graph structures quantitatively, following You et al. (2018b), we report distance metrics for node degree distribution, clustering coefficient distribution, and four-node orbit count distribution. We adopt 1-Wasserstein distance $W_1(x, y)$, where $x, y$ are respectively a graph statistic distribution from the original graph and a generated graph, and a lower value is better. Besides distribution distance metrics, we directly compare a few scalar-valued statistics. Let $M(G)$ be a non-negative-valued statistic, we report $\mathbb{E}_{\hat{G} \sim p_\theta}\left[M(\hat{G})/M(G)\right]$, where $\hat{G}$ is a generated graph. A value closer to 1 is better. In addition to triangle count, we employ the following metric for measuring the correlations between graph structure and node label (Lim et al., 2021), where a larger value indicates a higher correlation. $\hat{h}(\mathbf{A}, \mathbf{Y}) = \frac{1}{C_Y - 1} \sum_{k=1}^{C_Y} \max\left\{0, \frac{\sum_{\mathbf{Y}_v=k} |\{u \in \mathcal{N}(v)|\mathbf{Y}_u = \mathbf{Y}_v\}|}{\sum_{\mathbf{Y}_v=k} |\mathcal{N}(v)|} - \frac{|\{v|\mathbf{Y}_v=k\}|}{N}\right\}$, where $\mathcal{N}(v)$ consists of the neighboring nodes of $v$. We also report the metric for two-hop correlations, denoted by $\hat{h}(\mathbf{A}^2, \mathbf{Y})$. The computation of clustering coefficients, orbit counts, and triangle count may be costly for the large generated graphs. So, in such cases, we sample an edge-induced subgraph with the same number of edges from all generated graphs.

Table 1 displays the evaluation results for Cora and Amazon Computer. For results on Citeseer and Amazon Photo, see Appendix C. Unless otherwise mentioned, we report all results in this paper

Table 1: Evaluation with statistics on Cora and Amazon Computer. Best results are in **bold**.

| Model | Cora | | | | | | Amazon Computer | | | | | |
|---|---|---|---|---|---|---|---|---|---|---|---|---|
| | $W_1 \downarrow$ | | | $\mathbb{E}_{\hat{G} \sim p_\theta}\left[M(\hat{G})/M(G)\right] \to 1$ | | | $W_1 \downarrow$ | | | $\mathbb{E}_{\hat{G} \sim p_\theta}\left[M(\hat{G})/M(G)\right] \to 1$ | | |
| | Degree | Cluster | Orbit | # Triangle | $\hat{h}(\mathbf{A}, \mathbf{Y})$ | $\hat{h}(\mathbf{A}^2, \mathbf{Y})$ | Degree | Cluster | Orbit | # Triangle | $\hat{h}(\mathbf{A}, \mathbf{Y})$ | $\hat{h}(\mathbf{A}^2, \mathbf{Y})$ |
| ER | 1.0 | $2.4e^1$ | 1.6 | $6.1e^{-3}$ | $9.4e^{-3}$ | $9.7e^{-2}$ | $2.5e^1$ | $3.1e^{-1}$ | 2.1 | $9.7e^{-3}$ | $9.2e^{-4}$ | $2.9e^{-3}$ |
| SBM | $9.6e^{-1}$ | $2.3e^1$ | 1.6 | $2.4e^{-2}$ | **1.0** | **1.0** | **$2.1e^1$** | $3.0e^{-1}$ | 2.0 | $1.8e^{-2}$ | **1.0** | 1.6 |
| Feature-based MF | $1.3e^3$ | $2.4e^1$ | 1.6 | $4.9e^{-3}$ | $1.1e^{-1}$ | 0.0 | $6.8e^3$ | $3.1e^{-1}$ | 2.1 | $3.6e^{-3}$ | $6.7e^{-2}$ | 0.0 |
| GAE | $1.3e^3$ | $2.4e^1$ | 1.6 | $7.1e^{-3}$ | $1.1e^{-1}$ | 0.0 | $7.5e^3$ | $3.2e^{-1}$ | 2.1 | 0.0 | $3.4e^{-2}$ | 0.0 |
| VGAE | $1.4e^3$ | $2.4e^1$ | 1.6 | $6.8e^{-3}$ | $1.0e^{-1}$ | 0.0 | $6.8e^3$ | $3.2e^{-1}$ | 2.1 | 0.0 | $6.7e^{-2}$ | 0.0 |
| GraphMaker-Sync | **$6.2e^{-1}$** | $2.3e^1$ | 1.3 | $7.1e^{-2}$ | $9.2e^{-1}$ | $9.5e^{-1}$ | $2.6e^2$ | $2.9e^{-1}$ | 1.5 | $6.6e^{-2}$ | 1.2 | $5.3e^{-1}$ |
| GraphMaker-Async | 1.5 | **9.1** | $6.1e^{-1}$ | 1.4 | 1.1 | 1.1 | $2.1e^2$ | $2.0e^{-1}$ | 1.2 | **$4.2e^{-1}$** | 1.3 | **1.2** |
| GraphMaker-E | 1.0 | $1.0e^1$ | **$4.2e^{-1}$** | 1.4 | 1.1 | 1.2 | $2.6e^1$ | **$1.2e^{-1}$** | **$8.4e^{-1}$** | 1.9 | 1.1 | 2.5 |

Table 2: Evaluation with discriminative models on Cora and Amazon Computer. Best results are in **bold**. Highest results are underlined.

| Model | Cora | | | | | | | | Amazon Computer | | | | | | | |
|---|---|---|---|---|---|---|---|---|---|---|---|---|---|---|---|---|
| | Node Classification $\to 1$ | | | | | Link Prediction $\to 1$ | | | Node Classification $\to 1$ | | | | | Link Prediction $\to 1$ | | |
| | 1-SGC | L-SGC | L-GCN | 1-APPNP | L-APPNP | CN | 1-GAE | L-GAE | 1-SGC | L-SGC | L-GCN | 1-APPNP | L-APPNP | CN | 1-GAE | L-GAE |
| ER | 0.53 | 0.75 | 0.73 | 0.83 | 0.90 | 0.79 | 0.90 | 0.77 | 0.27 | 0.39 | 0.19 | 0.75 | 0.71 | 0.85 | 0.96 | 0.85 |
| SBM | 1.01 | 1.06 | 1.02 | 1.03 | 1.04 | **1.00** | **0.98** | 0.98 | 0.89 | 0.81 | 0.91 | **0.95** | 0.97 | **1.00** | 0.97 | **1.00** |
| Feature-based MF | 1.05 | 0.75 | 0.65 | 1.04 | 1.01 | 0.70 | 0.96 | 0.72 | 0.20 | 0.11 | 0.10 | 0.90 | 0.91 | 0.54 | **0.99** | 0.84 |
| GAE | 1.05 | 0.85 | 0.80 | 1.05 | 1.01 | 0.70 | 0.96 | 0.74 | 0.44 | 0.30 | 0.35 | 0.86 | 0.68 | 0.54 | 0.98 | 0.87 |
| VGAE | 1.01 | 0.63 | 0.47 | 1.03 | **1.00** | 0.70 | 0.97 | 0.74 | 0.29 | 0.22 | 0.13 | 0.90 | 0.88 | 0.54 | **0.99** | 0.80 |
| GraphMaker-Sync | 0.93 | 1.01 | 1.01 | **0.99** | 1.01 | **1.00** | **0.98** | 0.98 | 0.85 | 0.89 | 0.92 | 0.91 | 0.94 | 0.85 | 0.98 | 0.99 |
| GraphMaker-Async | 0.93 | **1.00** | **1.00** | 0.96 | 1.01 | **1.00** | **0.98** | **1.00** | **1.00** | **0.98** | 0.96 | 0.94 | 0.99 | 0.82 | 0.97 | 1.01 |
| GraphMaker-E | **1.00** | 1.05 | 1.05 | 1.04 | 1.05 | **1.00** | **0.98** | 0.99 | 0.96 | 0.96 | **0.97** | 0.94 | **1.00** | **1.00** | **0.99** | 1.01 |

based on 10 generated graphs that have the same number of nodes as the original graph. Out of 24 metrics, GraphMaker variants achieve the best performance for 16 of them. The best GraphMaker variants for each dataset collectively yield the best performance on 12 metrics. SBM is the most competitive baseline. GraphMaker-E performs better than SBM for 13 metrics. GraphMaker-Async indeed addresses the problem mentioned in Section 3.2, consistently surpassing GraphMaker-Sync for clustering coefficient distribution and triangle count. Overall, it performs better than or comparable to GraphMaker-Sync for 16/24 cases, which demonstrates the benefits of disentangling the generation of node attributes and graph structure in certain cases.

## 4.3 EVALUATION WITH DISCRIMINATIVE MODELS

To evaluate the utility of the generated graphs for downstream ML tasks, we introduce a novel evaluation protocol based on discriminative models. We train one model on the training set of the original graph $(G, \mathbf{Y})$ and another model on the training set of a generated graph $(\hat{G}, \hat{\mathbf{Y}})$. We then evaluate the two models on the test set of the original graph to obtain two performance metrics $\text{ACC}(G|G)$ and $\text{ACC}(G|\hat{G})$. If the ratio $\text{ACC}(G|\hat{G})/\text{ACC}(G|G)$ is close to one, then the generated graph is considered as having a utility similar to the original graph for training the particular model. We properly tune each model to ensure a fair comparison, see Appendix D for more details.

**Node classification.** Node classification models evaluate the correlations between unlabeled graphs and their corresponding node labels. Given our focus on the scenario with a single large graph, we approach the semi-supervised node classification problem. We randomly split a generated dataset so that the number of labeled nodes in each class and each subset is the same as that in the original dataset . For discriminative models, we choose three representative GNNs – SGC (Wu et al., 2019), GCN (Kipf & Welling, 2017), and APPNP (Gasteiger et al., 2019). As they employ different orders for message passing and prediction, this combination allows for examining data patterns more comprehensively. To scrutinize the retention of higher-order patterns, we employ two variants for each model, one with a single message passing layer denoted by $1-*$ and another with multiple message passing layers denoted by $L-*$. In addition, we directly evaluate the correlations between node attributes and node label with MLP in appendix D.

**Link prediction.** This task requires predicting missing edges in an incomplete graph, potentially utilizing node attributes and node labels. To prevent label leakage from dataset splitting, we split the edges corresponding to the upper triangular adjacency matrix into different subsets and then add reverse edges after the split. Following the practice of Kipf & Welling (2016), we adopt ROC-AUC as the evaluation metric. We consider two types of discriminative models – CN (Liben-Nowell & Kleinberg, 2003) and GAE (Kipf & Welling, 2016). CN is a traditional method that predicts edge

Table 3: Evaluation for benchmarking ML models. Best results are in **bold**.

| | Cora | | Citeseer | | Amazon Photo | | Amazon Computer | |
|---|---|---|---|---|---|---|---|---|
| Model | Pearson ↑ | Spearman ↑ | Pearson ↑ | Spearman ↑ | Pearson ↑ | Spearman ↑ | Pearson ↑ | Spearman ↑ |
| ER | -0.88 | -0.08 | -0.81 | -0.07 | 0.25 | 0.46 | 0.09 | 0.07 |
| SBM | **0.99** | 0.91 | 0.94 | 0.77 | 0.52 | 0.57 | 0.76 | 0.71 |
| Feature-based MF | 0.01 | -0.25 | -0.15 | -0.21 | 0.22 | 0.42 | 0.33 | 0.46 |
| GAE | -0.01 | -0.21 | -0.21 | -0.15 | 0.41 | 0.44 | 0.22 | 0.45 |
| VGAE | 0.00 | -0.30 | -0.20 | -0.07 | 0.13 | 0.33 | 0.02 | 0.31 |
| GraphMaker-Sync | 0.95 | **0.98** | 0.97 | **0.89** | 0.84 | 0.89 | 0.88 | **0.83** |
| GraphMaker-Async | 0.98 | 0.96 | **0.98** | 0.87 | **0.95** | **0.97** | **0.98** | 0.80 |
| GraphMaker-E | **0.99** | 0.95 | 0.97 | 0.80 | 0.78 | 0.89 | 0.87 | 0.78 |

Table 4: Ablation study for conditional generation on label $\mathbf{Y}$ for Cora. Better results are in **bold**.

| Model | Conditional $\mathbf{Y}$ | $W_1 \downarrow$ | | | $\mathbb{E}_{\hat{G} \sim p_\theta}\left[M(\hat{G})/M(G)\right] \to 1$ | | | Node Classification $\to 1$ | | | | | | Link Prediction $\to 1$ | | |
|---|---|---|---|---|---|---|---|---|---|---|---|---|---|---|---|---|
| | | Degree | Cluster | Orbit | # Triangle | $\hat{h}(\mathbf{A}, \mathbf{Y})$ | $\hat{h}(\mathbf{A}^2, \mathbf{Y})$ | MLP | 1-SGC | L-SGC | L-GCN | 1-APPNP | L-APPNP | CN | 1-GAE | L-GAE |
| Sync | ✓ | **6.2e$^{-1}$** | **2.3e$^1$** | 1.3 | **7.1e$^{-2}$** | 9.2e$^{-1}$ | 9.5e$^{-1}$ | 1.00 | 0.93 | 1.01 | 1.01 | 0.99 | 1.01 | 1.00 | 0.98 | 0.98 |
| | | 1.2 | 2.4e$^1$ | 1.7 | 3.1e$^{-3}$ | 7.3e$^{-2}$ | 2.0e$^{-1}$ | 0.58 | 0.41 | 0.43 | 0.40 | 0.41 | 0.40 | 0.76 | 0.92 | 0.84 |
| Async | ✓ | **1.5** | **9.1** | 6.1e$^{-1}$ | 1.4 | 1.1 | 1.1 | 1.04 | 0.93 | 1.00 | 1.00 | 0.96 | 1.01 | 1.00 | 0.98 | 1.00 |
| | | 1.6 | 1.5e$^1$ | **5.6e$^{-1}$** | **8.2e$^{-1}$** | 8.6e$^{-1}$ | 7.3e$^{-1}$ | 0.89 | 0.89 | 0.97 | 1.01 | 0.94 | **1.00** | 1.00 | 0.98 | 0.99 |

existence if the number of common neighbors shared by two nodes exceeds a threshold. GAE is an MPNN-based model that integrates the information of graph structure, node attributes, and node labels. As before, we consider two variants for GAE.

Table 2 presents the results for node classification and link prediction on Cora and Amazon Computer. For the detailed results on Citeseer and Amazon Photo, see Appendix D. Out of 32 cases, GraphMaker variants achieve the best performance for 24 of them. The best GraphMaker variants for each dataset collectively yield the best performance for 18 cases. SBM still performs the best among the baselines and leads to the best performance for 8 cases. GraphMaker-E performs better than or comparable to SBM for 19 cases. Among GraphMaker variants, GraphMaker-Async overall performs the best and is better than or comparable to GraphMaker-Sync for $20/32$ cases, consistent with the result of statistics-based evaluation. Surprisingly, we find $\text{ACC}(G|\hat{G})$ (resp. $\text{AUC}(G|\hat{G})$) to be often greater than $\text{ACC}(G|G)$ (resp. $\text{AUC}(G|G)$), sometimes by a quite large margin up to about 0.2 as in the case of Citeseer. This observation suggests the potential of utilizing generative models to improve discriminative models for future work.

**Utility for Benchmarking ML Models on Node Classification.** Another important scenario of using generated graphs is benchmarking ML models, where industry practitioners aim to select the most effective model architecture from multiple candidates based on their performance on publicly available synthetic graphs and then train it from scratch on their proprietary real graph. This use case requires the generated graphs to yield relative performance of the candidate model architectures reproducible on the original graph. Following Yoon et al. (2023), for each candidate model architecture, we train and evaluate one model on the original graph for $\text{ACC}(G|G)$ and another on a generated graph for $\text{ACC}(\hat{G}|\hat{G})$. We then report the Pearson/Spearman correlation coefficients between them (Myers et al., 2010). Specifically, we include all six model architectures for node classification in the candidate set of model architectures. Table 3 presents the experiment results. Out of 8 cases, all GraphMaker variants outperform all baselines for 7 of them.

**Diversity.** Based on evaluation with statistics and discriminative models, we also show that Graph-Maker is capable of generating diverse and hence novel graphs in Appendix E.

## 4.4 Ablation Study for Denoising Node Labels

We perform an ablation study on if drawing $\mathbf{Y} \sim \mathbb{P}_{\mathcal{Y}}$ estimated from the original data for conditional generation yields better generation quality than treating node label as a node attribute and training GraphMaker to generate it with denoising. Table 4 presents the evaluation results on Cora with statistics and discriminative models. For more than $80\%$ cases, conditional generation yields a better performance. In particular, we observe significant performance improvement when evaluating GraphMaker-Sync with discriminative models by up to more than a half. This demonstrates the advantage of the conditional generation framework.

Table 5: Ablation study for learning a node-personalized model on Cora. Better results are in **bold**.

| Model | PE | $W_1 \downarrow$ | | | $\mathbb{E}_{\hat{G} \sim p_\#} \left[ M(\hat{G})/M(G) \right] \to 1$ | | |
|---|---|---|---|---|---|---|---|
| | | Degree | Cluster | Orbit | # Triangle | $\hat{h}(\mathbf{A}, \mathbf{Y})$ | $\hat{h}(\mathbf{A}^2, \mathbf{Y})$ |
| Sync | | $6.2e^{-1}$ | $\mathbf{2.3e^1}$ | **1.3** | $\mathbf{7.1e^{-2}}$ | $\mathbf{9.2e^{-1}}$ | $\mathbf{9.5e^{-1}}$ |
| Sync | ✓ | $\mathbf{5.3e^{-1}}$ | $\mathbf{2.3e^1}$ | 1.4 | $\mathbf{6.0e^{-2}}$ | $\mathbf{9.1e^{-1}}$ | $\mathbf{9.6e^{-1}}$ |
| Async | | 1.5 | 9.1 | $6.1e^{-1}$ | **1.4** | **1.1** | **1.1** |
| Async | ✓ | **1.2** | 7.6 | $\mathbf{4.9e^{-1}}$ | 1.6 | **1.1** | 1.2 |
| E | | **1.0** | $\mathbf{1.0e^1}$ | $\mathbf{4.2e^{-1}}$ | 1.4 | **1.1** | **1.2** |
| E | ✓ | 1.3 | $1.3e^1$ | $7.0e^{-1}$ | $\mathbf{9.7e^{-1}}$ | **1.1** | **1.2** |

| Model | PE | Node Classification $\to 1$ | | | | | | Link Prediction $\to 1$ | | |
|---|---|---|---|---|---|---|---|---|---|---|
| | | MLP | 1-SGC | L-SGC | L-GCN | 1-APPNP | L-APPNP | CN | 1-GAE | L-GAE |
| Sync | | **1.00** | **0.93** | 1.01 | **1.01** | **0.99** | 1.01 | **1.00** | **0.98** | **0.98** |
| Sync | ✓ | 1.02 | 0.89 | **0.99** | 0.99 | 0.95 | **1.00** | **1.00** | 0.97 | **0.98** |
| Async | | / | **0.93** | **1.00** | **1.00** | **0.96** | 1.01 | **1.00** | **0.98** | **1.00** |
| Async | ✓ | / | **0.93** | **1.00** | 0.99 | 0.95 | **1.00** | **1.00** | **0.98** | 0.99 |
| E | | / | **1.00** | 1.05 | **1.05** | **1.04** | **1.05** | **1.00** | **0.98** | **0.99** |
| E | ✓ | / | 1.01 | 1.06 | **1.05** | **1.04** | **1.05** | **1.00** | **0.98** | **0.99** |

## 4.5 ABLATION STUDY FOR NODE PERSONALIZATION

We perform an ablation study to understand if capturing each node's personalized behavior helps improve the graph generation quality. We adopt a state-of-the-art positional encoding (PE) method RFP (Eliasof et al., 2023). RFP uses random node features after a certain number of graph convolutions as PEs, where the random initial features can be viewed as some identity information to distinguish a node from the others. In our experiments, for each noisy graph either during training or generation, we compute PEs based on the current noisy graph structure and then use PEs as extra node attributes. This essentially assists the model in encoding more personalized behaviors of nodes. We examine the effects of such personalization on GraphMaker for Cora. Table 5 presents the evaluation results with statistics and discriminative models. For more than $80\%$ cases, GraphMaker variants without PE perform better than or comparable to GraphMaker variants with PE, which suggests that personalized node behaviors may not be really useful to learn a graph generative model, at least in our downstream evaluation.

## 5 FURTHER RELATED WORKS

We have reviewed the diffusion-based graph generative models in Section 1. Next, we review some other non-diffusion deep generative models and some recent improvement efforts on synthetic graph data evaluation. We leave a review on classical graph generative models to Appendix F.

**Non-diffusion deep generative models of large graphs.** GAE and VGAE (Kipf & Welling, 2016) extend AE and VAE (Kingma & Welling, 2014; Rezende et al., 2014) respectively for reconstructing and generating the structure of a large attributed graph. NetGAN (Bojchevski et al., 2018) extends WGAN (Arjovsky et al., 2017) for graph structure generation by sequentially generating random walks. GraphRNN (You et al., 2018b) and Li et al. (2018) propose auto-regressive models that generate graph structures by sequentially adding individual nodes and edges. GRAN (Liao et al., 2019) introduces a more efficient auto-regressive model that generates graph structures by progressively adding subgraphs. CGT (Yoon et al., 2023) considers a simplified version of large attributed graph generation. It clusters real node attributes during data pre-processing and generates subgraphs with a single categorical node label that indicates the attribute cluster.

**Evaluation of generated graphs with discriminative models.** GraphWorld (Palowitch et al., 2022) is a software that allows users to benchmark discriminative models on a large amount of synthetic attributed graphs, which can be generated by approaches like SBM, but it does not compare synthetic graphs against a real graph in benchmarking. CGT (Yoon et al., 2023) has studied using generated graphs to benchmark GNN models. With fixed model hyperparameters, it evaluates one model for node classification on the original graph and another on a generated graph, and then measures the performance correlation and discrepancy of the two models. Our work instead is the first attempt to train discriminative models on a generated graph and then evaluate them on the original graph.

## 6 CONCLUSION

We propose GraphMaker, a diffusion model capable of generating large attributed graphs, along with a novel evaluation protocol that assesses generation quality by training models on generated graphs and evaluating their performance on real graphs. Overall, GraphMaker achieves better performance compared to baselines, for both existing metrics and the newly proposed evaluation protocol. In the future, we plan to extend GraphMaker to generate even larger graphs with 1M+ nodes and more complex characteristics such as continuous-valued node attributes, node labels that indicate node anomalies for anomaly detection, etc.

## REPRODUCIBILITY STATEMENT

We submit the source code for GraphMaker as a supplementary material. After unzipping the downloaded file, the README file includes the instructions for setting up the environment, and training and evaluating GraphMaker models on all four datasets employed.

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

Table 6: Dataset statistics. For $|\mathcal{E}|$, we add reverse edges and then remove duplicate edges.

| Dataset | $|\mathcal{V}|$ | $|\mathcal{E}|$ | # labels | # attributes |
|---------|------|------|----------|--------------|
| Cora | $2,708$ | $10,556$ | 7 | $1,433$ |
| Citeseer | $3,327$ | $9,228$ | 6 | $3,703$ |
| Amazon Photo | $7,650$ | $238,163$ | 8 | 745 |
| Amazon Computer | $13,752$ | $491,722$ | 10 | 767 |

Duncan J. Watts and Steven H. Strogatz. Collective dynamics of 'small-world' networks. *Nature*, 393:440–442, 1998.

Felix Wu, Amauri Souza, Tianyi Zhang, Christopher Fifty, Tao Yu, and Kilian Weinberger. Simplifying graph convolutional networks. In *Proceedings of the 36th International Conference on Machine Learning*, pp. 6861–6871, 2019.

Keyulu Xu, Chengtao Li, Yonglong Tian, Tomohiro Sonobe, Ken-ichi Kawarabayashi, and Stefanie Jegelka. Representation learning on graphs with jumping knowledge networks. In *Proceedings of the 35th International Conference on Machine Learning*, pp. 5453–5462, 2018.

Minji Yoon, Yue Wu, John Palowitch, Bryan Perozzi, and Russ Salakhutdinov. Graph generative model for benchmarking graph neural networks. In *Proceedings of the 40th International Conference on Machine Learning*, pp. 40175–40198, 2023.

Jiaxuan You, Bowen Liu, Zhitao Ying, Vijay Pande, and Jure Leskovec. Graph convolutional policy network for goal-directed molecular graph generation. In *Advances in Neural Information Processing Systems*, 2018a.

Jiaxuan You, Rex Ying, Xiang Ren, William Hamilton, and Jure Leskovec. GraphRNN: Generating realistic graphs with deep auto-regressive models. In *Proceedings of the 35th International Conference on Machine Learning*, pp. 5708–5717, 2018b.

Yunpeng Zhao, Elizaveta Levina, and Ji Zhu. Consistency of community detection in networks under degree-corrected stochastic block models. *The Annals of Statistics*, 40(4):2266, 2012.

## A  JUSTIFICATION FOR RECONSTRUCTING THE ORIGINAL GRAPH

By the Bayes rule, we have $q(\mathbf{X}_{v,f}^{(t-1)}|G^{(t)}, t, G^{(0)}) \propto \mathbf{X}_{v,f}^{(t)}(\mathbf{Q}_{X_f}^{(t)})^\top \odot \mathbf{X}_{v,f}\bar{\mathbf{Q}}_{X_f}^{(t-1)}$, where $\mathbf{Q}_{X_f}^{(t)} = (\bar{\mathbf{Q}}_{X_f}^{(t-1)})^{-1}\bar{\mathbf{Q}}_{X_f}^{(t)}$, and $\odot$ is the elementwise product. Similarly, we have $q(\mathbf{A}_{u,v}^{(t-1)}|G^{(t)}, t, G^{(0)}) \propto \mathbf{A}_{u,v}^{(t)}(\mathbf{Q}_A^{(t)})^\top \odot \mathbf{A}_{u,v}\bar{\mathbf{Q}}_A^{(t-1)}$.

## B  DATASET STATISTICS

Table 6 presents the detailed dataset statistics.

## C  ADDITIONAL RESULTS FOR EVALUATION WITH STATISTICS

Table 7 presents the detailed results of evaluation with statistics for Citeseer and Amazon Photo.

## D  ADDITIONAL DETAILS FOR EVALUATION WITH DISCRIMINATIVE MODELS

We design a hyperparameter space specific to each discriminative model, and implement a simple AutoML pipeline to exhaustively search through a hyperparameter space for the best trained model.

Table 8 presents the results for node attribute classification with MLP. The conditional empirical distribution $p(\mathbf{Y})\prod_v \prod_f p(\mathbf{X}_{v,f}|\mathbf{Y}_v)$ consistently outperforms the unconditional

Table 7: Evaluation with statistics on Citeseer and Amazon Photo. Best results are in **bold**.

| Model | | Citeseer | | | | | | Amazon Photo | | | | |
|---|---|---|---|---|---|---|---|---|---|---|---|---|
| | $W_1 \downarrow$ | | | $\mathbb{E}_{\hat{G} \sim p_\theta}\left[M(\hat{G})/M(G)\right] \to 1$ | | | $W_1 \downarrow$ | | | $\mathbb{E}_{\hat{G} \sim p_\theta}\left[M(\hat{G})/M(G)\right] \to 1$ | | |
| | Degree | Cluster | Orbit | # Triangle | $\hat{h}(\mathbf{A}, \mathbf{Y})$ | $\hat{h}(\mathbf{A}^2, \mathbf{Y})$ | Degree | Cluster | Orbit | # Triangle | $\hat{h}(\mathbf{A}, \mathbf{Y})$ | $\hat{h}(\mathbf{A}^2, \mathbf{Y})$ |
| ER | $8.5e^{-1}$ | $1.4e^1$ | $1.6$ | $3.5e^{-3}$ | $9.0e^{-3}$ | $1.8e^{-1}$ | $1.9e^1$ | $4.0e^1$ | $1.5$ | $7.0e^{-3}$ | $1.9e^{-3}$ | $2.2e^{-3}$ |
| SBM | $\mathbf{8.0e^{-1}}$ | $1.4e^1$ | $1.5$ | $8.3e^{-3}$ | $\mathbf{1.0}$ | $8.1e^{-1}$ | $1.5e^1$ | $3.8e^1$ | $1.4$ | $3.9e^{-2}$ | $\mathbf{1.0}$ | $\mathbf{1.2}$ |
| Feature-based MF | $1.7e^3$ | $1.4e^1$ | $1.6$ | $3.9e^{-3}$ | $1.1e^{-1}$ | $0.0$ | $3.8e^3$ | $4.0e^1$ | $1.5$ | $7.2e^{-3}$ | $1.0e^{-1}$ | $0.0$ |
| GAE | $1.7e^3$ | $1.4e^1$ | $1.6$ | $2.0e^{-3}$ | $7.7e^{-2}$ | $0.0$ | $3.8e^3$ | $4.0e^1$ | $1.5$ | $7.2e^{-3}$ | $6.2e^{-2}$ | $0.0$ |
| VGAE | $1.7e^3$ | $1.4e^1$ | $1.6$ | $3.2e^{-3}$ | $8.5e^{-2}$ | $0.0$ | $3.8e^3$ | $4.0e^1$ | $1.5$ | $7.2e^{-3}$ | $1.2e^{-1}$ | $0.0$ |
| GraphMaker-Sync | $1.1$ | $1.4e^1$ | $1.2$ | $3.6e^{-2}$ | $9.3e^{-1}$ | $7.9e^{-1}$ | $7.6e^1$ | $3.7e^1$ | $1.2$ | $1.0e^{-1}$ | $1.1$ | $7.6e^{-1}$ |
| GraphMaker-Async | $9.3$ | $1.3e^1$ | $8.8e^{-1}$ | $2.6e^{-1}$ | $1.4$ | $1.4$ | $\mathbf{9.7}$ | $\mathbf{1.9e^1}$ | $\mathbf{4.3e^{-1}}$ | $\mathbf{4.4e^{-1}}$ | $1.1$ | $1.4$ |
| GraphMaker-E | $6.8$ | $\mathbf{1.0e^1}$ | $\mathbf{6.8e^{-1}}$ | $\mathbf{4.8e^{-1}}$ | $1.2$ | $\mathbf{1.1}$ | $1.6e^1$ | $2.6e^1$ | $6.8e^{-1}$ | $2.4e^{-1}$ | $1.1$ | $1.5$ |

Table 8: Evaluation with MLP. Best results are in **bold**. Highest results are underlined.

| Model | Cora | Citeseer | Amazon Photo | Amazon Computer |
|---|---|---|---|---|
| $p(\mathbf{Y}) \prod_v \prod_f p(\mathbf{X}_{v,f})$ | 0.58 | 0.35 | 0.16 | 0.11 |
| $p(\mathbf{Y}) \prod_v \prod_f p(\mathbf{X}_{v,f}\vert\mathbf{Y}_v)$ | 1.10 | 1.23 | **0.99** | **0.97** |
| GraphMaker-Sync | **1.00** | 1.12 | 0.97 | 0.94 |
| GraphMaker-Async | 1.04 | **1.10** | 1.04 | **0.97** |

variant $p(\mathbf{Y}) \prod_v \prod_f p(\mathbf{X}_{v,f})$. On three datasets, GraphMaker variants are better than $p(\mathbf{Y}) \prod_v \prod_f p(\mathbf{X}_{v,f}\vert\mathbf{Y}_v)$ in capturing the correlations between node attributes and node label. Table 9 and 10 present the detailed evaluation results with discriminative models on Citeseer and Amazon Photo.

## E EVALUATION FOR DIVERSITY AND NOVELTY

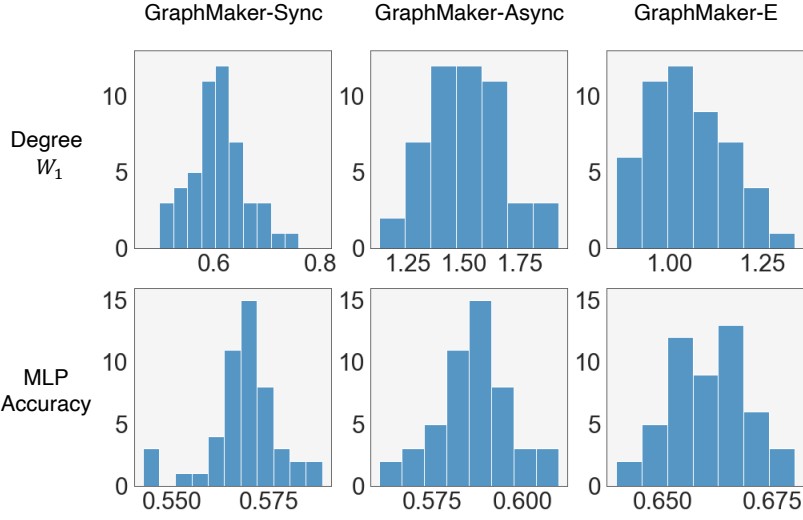

Figure 2: Histogram plots of metrics, which demonstrate the diversity of the generated graphs.

To study the diversity of the graphs generated by GraphMaker, we generate 50 graphs with each GraphMaker variant and make histogram plots of metrics based on them. For structure diversity, we report $W_1$ for node degree distribution. For node attribute and label diversity, we train an MLP on the original graph and report its accuracy on generated graphs. Figure 2 presents the histogram plots for Cora, which demonstrates that GraphMaker is capable of generating diverse and hence novel graphs.

Table 9: Evaluation with discriminative models on Citeseer. Best results are in **bold**. Highest results are underlined.

| Model | Node Classification → 1 | | | | | Link Prediction → 1 | | |
|---|---|---|---|---|---|---|---|---|
| | 1-SGC | L-SGC | L-GCN | 1-APPNP | L-APPNP | CN | 1-GAE | L-GAE |
| ER | 0.81 | 0.85 | **0.96** | **0.96** | **0.99** | 0.90 | 0.93 | 0.80 |
| SBM | 1.13 | 1.13 | 1.12 | 1.12 | 1.15 | 0.90 | 0.99 | 0.98 |
| Feature-based MF | 1.20 | **0.96** | 0.67 | 1.16 | 1.16 | 0.75 | 0.97 | 0.75 |
| GAE | 1.22 | 0.67 | 0.29 | 1.16 | 1.16 | 0.75 | 0.97 | 0.75 |
| VGAE | 1.20 | 0.79 | 0.52 | 1.15 | 1.15 | 0.75 | 0.97 | 0.76 |
| GraphMaker-Sync | **1.05** | 1.06 | 1.06 | **1.04** | 1.09 | **1.00** | 0.98 | 0.98 |
| GraphMaker-Async | 1.13 | 1.12 | 1.12 | 1.07 | 1.13 | **1.00** | 0.97 | 0.97 |
| GraphMaker-E | 1.19 | 1.16 | 1.14 | 1.15 | 1.18 | **1.00** | 0.98 | **0.99** |

Table 10: Evaluation with discriminative models on Amazon Photo. Best results are in **bold**. Highest results are underlined.

| Model | Node Classification → 1 | | | | | Link Prediction → 1 | | |
|---|---|---|---|---|---|---|---|---|
| | 1-SGC | L-SGC | L-GCN | 1-APPNP | L-APPNP | CN | 1-GAE | L-GAE |
| ER | 0.34 | 0.59 | 0.15 | 0.73 | 0.66 | 0.90 | 0.94 | 0.67 |
| SBM | 1.05 | **1.00** | 0.82 | 0.82 | 0.81 | **1.00** | 0.99 | **0.99** |
| Feature-based MF | 0.77 | 0.22 | 0.15 | 0.84 | 0.73 | 0.54 | 0.99 | 0.81 |
| GAE | 0.77 | 0.22 | 0.16 | 0.82 | 0.77 | 0.54 | 0.99 | 0.75 |
| VGAE | 0.77 | 0.30 | 0.20 | 0.83 | 0.79 | 0.54 | 0.99 | 0.77 |
| GraphMaker-Sync | **1.00** | 0.97 | 0.89 | 0.88 | 0.90 | 0.99 | 0.98 | 0.98 |
| GraphMaker-Async | 1.11 | 1.06 | **0.94** | **0.92** | **0.93** | **1.00** | **1.00** | **0.99** |
| GraphMaker-E | **1.00** | 0.92 | 0.87 | 0.88 | 0.88 | **1.00** | 0.99 | 0.98 |

# F REVIEW ON CLASSIC GRAPH GENERATIVE MODELS

ER (Erdős & Rényi, 1959) generates graph structures with a desired number of nodes and average node degree. SBM (Holland et al., 1983) produces graph structures with a categorical cluster label per node that meet target inter-cluster and intra-cluster edge densities. BA (Barabási & Albert, 2002) generates graph structures whose node degree distribution follows a power law. Chung-Lu (Chung & Lu, 2002) generates graphs with a node degree distribution that equals to a pre-specified node degree distribution in expectation. Kronecker graph model (Leskovec et al., 2010) generates realistic graphs recursively by iterating the Kronecker product.

