# OpenReview forum: "GraphMaker: Can Diffusion Models Generate Large Attributed Graphs?"
_ICLR.cc/2024/Conference — Submitted to ICLR 2024_

### Official Review · Reviewer_4wGq · 2023-10-24

**Soundness:** 2 fair
**Presentation:** 3 good
**Contribution:** 2 fair
**Rating:** 5
**Confidence:** 5

**Summary:**

This paper proposes a diffusion-based graph generative model which aims to learn a distribution of a *large, attributed* graph from a single sample. This differs from past work in that other diffusion-based graph generative models either learn (1) a distribution of a small, attributed  graph from many samples (such as molecules) or (2) a distribution of a large unattributed graph from a single sample. The authors argue that learning a large graph from a single sample is plausible because the nodes are statistically exchangable, thus the joint distribution of nodes can be effectively learned in this setting (logic which also underlies classical graph statistical models such as the SBM)

The authors propose GraphMaker, a denoising model learned from a sequence of data corruptions that are applied to the adjacency matrix and attribute matrix. GraphMaker-Sync denoises the adjacencies and attributes simultaneously, while GraphMaker-Async denoises the adjacencies after the attributes have been denoised.

The authors propose three evaluation aspects for their approach and competitors:

1) Graph property: how well do graph/attribute measures computed on the generated graph match the same measures on the source data

2) Discriminative: how well does a GNN trained on the generated graph perform on the source data

3) Benchmarking: how closely does the *rank* of GNN models benchmarked on the generated graph compare to the same ranks on the source data.

**Strengths:**

The main strength of this paper is that there are few graph generative models that can learn attributed-graph distributions in the large-graph setting. The only existing one (that I know of, which the authors cite) is Yoon et al. 2023 and appears to not actually generate the whole graph, but rather only batch-level training samples of rooted trees on which GNNs can be trained effectively. The fact that GraphMaker can generate a real graph sample with attributes at the same scale as the input data makes it a valuable contribution to the community.

**Weaknesses:**

The main weakness of the paper is that the empirical results show that the proposed method only marginally outperforms the SBM in the evaluation aspects (graph property, discriminative, benchmarking). While the GraphMaker graphs seem to match graph statistics slightly better (in aggregate), and also better align ranks in benchmarking, the discriminative aspect (Table 2) shows that GNN models trained on synthetic graphs from GraphMaker vs those from SBM do about the same when trained on the source data.

The fact that SBM is a powerful baseline is interesting, and could be due to a number of factors:

(1) the graphs used in the empirical study are too homophilous, and thus the label-conditioning aspect of GraphMaker could essentially be copying what the SBM is already doing.

(2) GraphMaker's noise injection is uniform over all edges / attribute dimensions: this could be too simple to go beyond the i.i.d. edge generation of SBM.

A related weakness is that the authors did not benchmark against the EDGE graph diffusion model (Chen et al. 2023), which would be an interesting baseline alongside the SBM, which similarly does not generate the attributes.

I elaborate on the above in my questions to the authors.

**Questions:**

Q1: Did you consider benchmarking on datasets that have less edge/attribute-level homophily? We should expect the performance of SBM to go down in this case, but potentially not for GraphMaker.

Q2: Why not include EDGE + attribute diffusion as a baseline, as you did for SBM and other graph-only models? I think the relative contribution of this work is hard to assess without this comparison.

Q3: From the description in Section 3.5, I cannot understand what GraphMaker-E is doing, and how it differs from GraphMaker-Async. Can you elaborate?

I am willing to raise my score if the authors can provide more info along these lines.

---

> ### Author Response · Authors · 2023-11-17
> **Response to Reviewer 4wGq**
>
> We highly appreciate your detailed and insightful feedback. We address the individual questions as follows.
>
> > 1. The manuscript presents an evaluation that examines how well models trained on the generated training data align with the models trained on the original training data, when applied to the original test data. The evaluation result shows that stochastic block model (SBM), when equipped with the empirical distribution computed from the original data $P(Y)\prod_v\prod_f P(X_{v, f} | Y_v)$, is a strong baseline. The proposed GraphMaker models only marginally outperform SBM. One possible explanation is that the graphs used in the empirical study are too homophilous, and thus the label-conditioning mechanism of GraphMaker could be essentially learning inter-cluster and intra-cluster edge densities employed by SBM. A natural question is then if the authors considered benchmarking on less homophilous graphs, where GraphMaker is more likely to have an edge in performance.
>
> This is indeed a critical insight. Various homophily measures have been developed to measure the similarity between nodes and their neighbors in a graph [1-3], in particular the proportion of nodes that have the same label as their neighbors. These papers have pointed out that Cora and Citeseer are highly homophilous, and it’s likely that Amazon Photo and Amazon Computer also have this issue. When we apply SBM to such graphs, with high probability the generated nodes will be connected to neighbors that have the same label. In addition, the node attributes can also be approximately conditionally independent of each other given the node label, as shown in Table 8 in the Appendix. These two factors may explain the competitive performance achieved by SBM for MPNNs, which perform localized smoothing.
>
> We agree that more heterophilous datasets with more complicated attribute, label, and structure correlations can be better testbeds. We plan to include such datasets in the next version of the paper.
>
> > 2. Lack of performance comparison against recent diffusion-based graph generative models like EDGE
>
> As mentioned in the above unified response, we agree that a direct comparison will help us better understand the strengths and limitations of the proposed model. We plan to include some previous diffusion-based graph generative models as additional baselines in the next version of the manuscript.
>
> > 3. What is GraphMaker-E doing? How does it differ from GraphMaker-Async?
>
> We apologize for not well articulating the presentation for this variant. As elaborated in the above unified response, we replace the trained attribute diffusion model with $P(Y)\prod_v\prod_f P(X_{v, f} | Y_v)$, the empirical distributions computed from the original data, as explained in section 4.1. This allows a direct comparison of the structure generation capability of GraphMaker against baselines without attribute generation capability.
>
> [1] Pei et al. Geom-GCN: Geometric Graph Convolutional Networks.
>
> [2] Zhu et al. Beyond Homophily in Graph Neural Networks: Current Limitations and Effective Designs.
>
> [3] Lim et al. Large Scale Learning on Non-Homophilous Graphs: New Benchmarks and Strong Simple Methods.

---

> > ### Comment · Reviewer_4wGq · 2023-11-23
> >
> > Thanks for the time taken in your response. I would like to see the proposed edits to the paper actually implemented before raising my score. Please feel free to upload a revision.

---

> > > ### Author Response · Authors · 2023-11-23
> > > **Further Response to Reviewer 4wGq**
> > >
> > > Thank you for your time in reading the response and kindly considering raising the score! Based on the overall feedback received from all reviewers, we don't think we can address them well before the end of the discussion period and we plan to improve the manuscript for a resubmission to another venue.

---

### Official Review · Reviewer_muBL · 2023-11-01

**Soundness:** 2 fair
**Presentation:** 2 fair
**Contribution:** 2 fair
**Rating:** 3
**Confidence:** 4

**Summary:**

This paper presents GraphMaker, a diffusion model for generating large attributed graphs. They presented three types of diffusion models that couple or decouple graph structure and node attribute generation, and that utilizes label conditions. Also, they present a new evaluation pipeline for graph generation.

**Strengths:**

1. A neat figure to describe the method GraphMaker-Sync and GraphMaker-Async, which helps the easier understanding of the method.
2. Presented a method for the generation of large attributed graphs given node labels.

**Weaknesses:**

1. The task that predicts the node labels and edge existence given the node attributes is more like a link prediction task, not a graph generation task. Is there any specific reason or reference that defines the task as a graph generation task?
2. What is the novelty of the proposed model? Diffusion-based graph generative models such as GDSS can also deal with attributed graphs. Also, for scalability, simple usage of MPNN for the encoder seems not to be a critical novelty point.
3. Hard to understand Section 3.4. What does it mean to learn from a single graph? Is it right that the model trains from only one graph and generates many graphs that fit with the training graph? When do we need such circumstances in the real world practically?
4. Lack of performance comparison for recent diffusion-based graph generative models such as GDSS, DiGress, and GraphARM. Need more recent baselines for the evaluation part.

**Questions:**

1. What about the cases with only one type of node attribute like molecular graphs? Do we treat node attributes and node labels to be the same or GraphMaker cannot deal with it?
2. What is the time complexity of GraphMaker? As the model is diffusion-based, it seems that the inference and training time may be long. How does GraphMaker benefit from large attributed graph generation? Hard to understand what a minibatch strategy is which is introduced in the introduction part.

---

> ### Author Response · Authors · 2023-11-17
> **Response to Reviewer muBL**
>
> Again, we thank the reviewer for the efforts in providing valuable feedback. We address the individual questions as follows.
>
> > 1. What’s the difference between link prediction and graph generation?
>
> Link prediction considers the scenario of inferring additional edges from an existing incomplete graph [1, 2]. In contrast, graph generation requires creating a graph from scratch without leveraging an existing incomplete graph [3, 4].
>
> > 2. What is the time complexity of GraphMaker for edge generation? What is the minibatch strategy introduced in the introduction part?
>
> GraphMaker first samples a noisy graph from the prior distribution, and then repeatedly refines it. At each timestep, it computes node representations with an MPNN, and then performs binary classification for all node pairs. This yields a time complexity of $O(TN^2)$, where $T$ is the number of diffusion steps and $N$ is the number of nodes. While this appears to be costly, we only need to perform node representation computation once per diffusion step, which can be used for all node pairs. Besides, binary classification for multiple node pairs given node representations can be performed in parallel. In addition, the optimal $T$ is less than 10 based on empirical studies. In terms of wall clock time, generating the largest graph (13K nodes) considered in the paper with the most costly GraphMaker model takes less than 10 minutes on an NVIDIA RTX A6000 GPU during inference.
>
> Another issue is the space complexity of $O(N^2)$, which prevents us from performing binary classification for all node pairs at once on a single GPU. Hence we adopt the minibatch strategy. During training, we use a minibatch of node pairs for a gradient update. During inference, we generate the whole graph structure by iterating over minibatches of node pairs.
>
> For larger graphs, we agree that $O(N^2)$ indeed will be problematic, and we may leverage methods for approximate nearest neighbor. For example, locality sensitive hashing [5] hashes similar data points into the same bucket with high probability, and therefore it avoids all pair comparisons.
>
> > 3. Lack of performance comparison against recent diffusion-based graph generative models
>
> As mentioned in the above unified response, we agree that a direct comparison will help us better understand the strengths and limitations of the proposed model. We plan to include some previous diffusion-based graph generative models as additional baselines in the next version of the manuscript.
>
> > 4. What are the application scenarios of large attributed graph generation?
>
> As mentioned in the above unified response, this task can benefit the understanding of attributed complex networks for network science and the sharing of sensitive networked data for public usage.
>
> [1] Liben-Nowell & Kleinberg. The Link Prediction Problem for Social Networks.
>
> [2] Zhang & Chen. Link Prediction Based on Graph Neural Networks.
>
> [3] You et al. GraphRNN: Generating Realistic Graphs with Deep Auto-regressive Models.
>
> [4] Yoon et al. Graph Generative Model for Benchmarking Graph Neural Networks.
>
> [5] Indyk & Motwani. Approximate nearest neighbors: towards removing the curse of dimensionality.

---

> > ### Comment · Reviewer_muBL · 2023-11-20
> >
> > Thank you for the kind response.
> >
> >  Despite your kind responses, I still cannot fully understand the novelty of GraphMaker and its superiority compared to other diffusion-based graph generative models. For instance, the authors have not answered my question about the novelty of GraphMaker as other baselines such as GDSS generate attributed graphs with diffusion models. In addition, the authors simply answer that “While these models may have various limitations for handling large attributed graphs” without providing any time complexity analysis or performance analysis compared to other diffusion-based models.
> >
> >  Moreover, I still cannot understand the exact setting that learns from a single graph. This seems similar to the setting of GVAE which provided the link prediction task as the node attributes are given and predicts the node labels and edge existence. However, the authors simply explain the difference between link prediction and graph generation, without enough explanation on why GraphMaker is targeting graph generation tasks.
> >
> > Therefore, I keep the score. Thank you for the detailed response again.

---

### Official Review · Reviewer_xCEF · 2023-11-03

**Soundness:** 2 fair
**Presentation:** 3 good
**Contribution:** 2 fair
**Rating:** 5
**Confidence:** 4

**Summary:**

This paper presents GraphMaker, a new diffusion model proposed for creating large attributed graphs. Large-scale graphs with node attributes are crucial in real-world contexts like social and financial networks. Generating synthetic graphs that mirror real ones is essential in graph machine learning, especially when original data is confidential. While traditional graph generation models have limitations, recent diffusion models excel in specific areas but face challenges with large attributed graphs. It explores diffusion models focusing on the relationship between graph structure and node attributes and introduces techniques for scalability. The paper also offers a different evaluation method using models trained on synthetic data. Experiments demonstrate GraphMaker's proficiency in producing realistic and diverse large-attributed graphs for subsequent applications.

--- Post rebuttal update --
The authors have partially addressed my questions, especially on applications and some technical details. Therefore, i raised my score from 3 to 5.

**Strengths:**

- S1. Generation of large attributed graph presents significant technical challenges that are worth investigation.

- S2. Writing is generally clear, although there are some clarity or motivational issues (see W1 and W2). But overall, it is easy to read and follow.

- S3. I like the evaluation methodology using ML models, which can be more expressive than traditional statistics, yet is general without requiring domain-specific knowledge.

**Weaknesses:**

- W1. Some technical details are not clearly introduced, especially in 3.5.

It was only mentioned that "which generates node attributes
through an external approach conditioned on node labels." How does this external approach/node label conditioning work exactly? What kind of label is suitable for this purpose? How are these node labels related to Y in 3.2?

These are not clearly explained.

- W2. Motivation of the decoupled approach is not well articulated, and the choice of the word "decoupled" is misleading.

First, the explanation/motivation of why GraphMaker-Asyn is better is not laid out convincingly. From the experiments, the results are also quite mixed. Essentially, together with GraphMaker-E, there are three alternative versions proposed, but there are no clear insight on why each version plays to its strength in certain scenarios.

Secondly, I think GraphMaker-Asyn is not really decoupled, as both edge and attribute generation are still trained together.

- W3. What is the difference between node attributes and labels? In Section 2, both are defined as categorical.

- W4. The application scenario of large attributed graph generation is unclear. Currently, generation of small graphs such as molecules are popular as they may potentially drive killer applications such as drug discovery.

**Questions:**

See weaknesses.

---

> ### Author Response · Authors · 2023-11-17
> **Response to Reviewer xCEF**
>
> Thank you so much for reviewing our paper and bringing valuable questions. We address the individual questions as follows.
>
> > 1. Motivation of the decoupled approach GraphMaker-Async is not well articulated. Besides, the choice of the word ``decoupled'' is misleading.
>
> We provided a motivation for choosing GraphMaker-Async over GraphMaker-Sync from the perspective of edge dependency modeling in the above unified response.
>
> We used the word ``decoupled'' as GraphMaker-Async consists of two subnetworks trained independently and once trained it first generates attributes and then edges conditioned on fixed attributes. We are open to other options if the reviewer has suggestions for better alternatives.
>
> > 2. What is the difference between node attributes and labels as both of them are categorical in section 2?
>
> The confusion comes from the fact that one important application of our generative model is to generate and publish synthetic graphs without sharing the original data. ML models (such as Graph Neural Networks) can be trained over these synthetic graphs and are expected to achieve similar performance as those trained directly over the original graphs. To evaluate our model for this application, we adopted node classification and link prediction tasks as a proof of concept and considered the large-attributed graphs that are widely used for node classification tasks [1, 2]. The node labels defined by the node classification tasks are used as the conditional information in our generation process. It is true that distinguishing labels and attributes is often just an artifact, defined by an ML task, which is irrelevant to the data generation procedure. However, if a downstream ML task is given, using the labels as conditions may indeed help. In our case, we find that directly viewing the node label as a node attribute yields worse performance for more than 80% cases for the downstream node classification tasks as shown in section 4.4.
>
> > 3. How are node attributes obtained for GraphMaker-E with an external approach conditioned on node labels?
>
> As explained in the unified response, we leverage the empirical distribution $P(Y)\prod_v\prod_f P(X_{v, f} | Y_v)$ computed from the original data for the experiments in this paper. We agree it’s better to mention it explicitly in 3.5 rather than 4.1.
>
> > 4. What is the application scenario of large attributed graph generation?
>
> As elaborated in the first paragraph and unified response, firstly, large attributed graph generation can enhance the understanding of complex networks [3, 4]. Secondly, the generated graphs can be shared with the public in replacement of the original sensitive data while preserving data utility.
>
> [1] Kipf et al. Semi-Supervised Classification with Graph Convolutional Networks.
>
> [2] Shchur et al. Pitfalls of Graph Neural Network Evaluation.
>
> [3] Watts & Strogatz. Collective dynamics of ‘small-world’ networks.
>
> [4] Barabási & Albert. Emergence of scaling in random networks.

---

> > ### Comment · Reviewer_xCEF · 2023-11-18
> > **Thanks for the response**
> >
> > Thanks for the detailed response. I will weigh them carefully in the final review.

---

> ### Comment · Reviewer_xCEF · 2023-12-03
>
> Based on the response (especially on the applications and some technical details), I have increased my score to 5.

---

### Author Response · Authors · 2023-11-17
**Unified Response to All Reviewers**

We thank the reviewers for reading our manuscript and providing constructive and insightful feedback. All reviewers agree that our paper is a valuable contribution and the presentation is good in general. In particular, the task of generating large attributed graphs presents significant technical challenges that are worth investigation (**Reviewer xCEF**), and GraphMaker is the first graph generative model that tackles this task (**Reviewer 4wGq**). Here, we first address three important questions shared by multiple reviewers in this unified response. We will further respond to each reviewer separately for the other questions.

> The first question is on the problem formulation and application scenario of large attributed graph generation (**Reviewer xCEF** and **muBL**).

As formulated in section 2, given a real-world large attributed graph, we are interested in learning its distribution. First, this allows understanding the evolution of complex attributed networks for network science. Traditional statistical random graph models [1] and deep generative models [2] have been developed for understanding complex network structures. Second, we can release the graphs sampled from the learned distribution in replacement of the original sensitive data [3, 4].

> The second question is on insufficient empirical studies and baselines, particularly the lack of performance comparison against other diffusion-based graph generative models, including DiGress [5], GDSS [6], EDGE [7], and GraphARM [8] (**Reviewer muBL** and **4wGq**).

As elaborated in section 3.4, DiGress adopts graph transformer [9] as the graph encoder, which has a complexity $O(N^2)$, where $N$ is the number of nodes, and is not scalable for large graphs. We have verified this point empirically. It takes a 16-GiB GPU for a single graph transformer layer, 4 attention heads and a hidden size of 16 per head, to compute over a graph with about two thousand nodes and one thousand attributes, let alone graphs with a larger scale. Other diffusion-based graph generative models were also initially developed for molecule or pure graph structure generation.

While these models may have various limitations for handling large attributed graphs, we agree that directly comparing GraphMaker against them in the form of empirical studies may provide a better and more straightforward understanding of their limitations, hence enhancing the argumentation made in the manuscript. We plan to include some of these models as additional baselines in the next version of the manuscript.

> The third question is on the differences and relative advantages of the three GraphMaker variants (GraphMaker-Sync, GraphMaker-Async, and GraphMaker-E) (**Reviewer xCEF** and **4wGq**).

GraphMaker-Sync is trained to simultaneously refine graph structure and attributes. GraphMaker-Async consists of two subnetworks trained independently. Once trained, it first generates node attributes, and then generates graph structure conditioned on the attributes. As shown in section 4.2, GraphMaker-Async consistently outperforms GraphMaker-Sync in capturing the distributions of triangle counts and clustering coefficients, demonstrating the advantage of async generation.

We agree that the motivation of asynchronous generation can be better articulated. [10] points out the limitations of edge independent graph generative models, models that generate edges independently, in capturing graph properties like triangle count. The fact that GraphMaker-Sync simultaneously refines attributes and structure makes it more difficult to capture edge dependencies. By contrast, GraphMaker-Async generates edges with multiple rounds of refinement based on fixed node attributes, which mitigates this issue.

GraphMaker-E is a special case of GraphMaker-Async, where we leverage the empirical distribution $P(Y)\prod_v\prod_f P(X_{v, f} | Y_v)$ from the original data, as explained in section 4.1. $Y$ is the label of all nodes. $X_{v,f}$ is the f-th attribute of the node $v$. This allows a direct comparison of the structure generation capability of GraphMaker against baselines without attribute generation capability.

[1] Barabási and Albert. Statistical mechanics of complex networks.

[2] You et al. GraphRNN: Generating Realistic Graphs with Deep Auto-regressive Models.

[3] Jorgensen et al. Publishing Attributed Social Graphs with Formal Privacy Guarantees.

[4] Yoon et al. Graph Generative Model for Benchmarking Graph Neural Networks.

[5] Vignac et al. DiGress: Discrete Denoising diffusion for graph generation.

[6] Jo et al. Score-based Generative Modeling of Graphs via the System of Stochastic Differential Equations.

[7] Chen et al. Efficient and Degree-Guided Graph Generation via Discrete Diffusion Modeling.

[8] Kong et al. Autoregressive diffusion model for graph generation.

[9] Dwivedi & Bresson. A Generalization of Transformer Networks to Graphs.

[10] Chanpuriya et al. On the Power of Edge Independent Graph Models.

---

### Meta-Review · Area_Chair_RMEj · 2023-12-03

**Metareview:**

This paper introduces GraphMaker, a novel diffusion-based graph generative model tailored for creating large attributed graphs. Addressing limitations in existing models, GraphMaker employs a denoising approach, simultaneously correcting adjacency and attribute matrices. The unique focus on learning a distribution of a large, attributed graph from a single sample distinguishes GraphMaker from prior diffusion-based models. The proposed evaluation framework assesses graph properties, discriminative performance, and benchmarking against source data. Experiments demonstrate GraphMaker's proficiency in generating realistic and diverse large-attributed graphs, showcasing its potential for applications in real-world scenarios.

The reviewer has noted a limited body of work on graph generation using Graph Neural Networks (GNN), and while there is a marginal improvement in performance, the authors did not offer additional clarification in response. Consequently, a significant revision of the paper is needed. I encourage the authors to thoroughly address the reviewer's comments, providing additional information where needed. It is recommended to undertake a comprehensive revision before resubmitting the paper to a suitable venue in the future.

**Justification For Why Not Higher Score:**

All reviewers rated this paper in either reject or weak reject, and I agree to their rating.

**Justification For Why Not Lower Score:**

N/A

---

### Decision · Program_Chairs · 2024-01-16

Reject